# Group and Shuffle: Efficient Structured Orthogonal Parametrization

**Mikhail Gorbunov**
HSE University
gorbunovmikh73@gmail.com

**Nikolay Yudin**
HSE University

**Vera Soboleva**
AIRI,
HSE University

**Aibek Alanov**
AIRI,
HSE University

**Alexey Naumov**
HSE University,
Steklov Mathematical Institute of Russian Academy of Sciences

**Maxim Rakhuba**
HSE University

## Abstract

The increasing size of neural networks has led to a growing demand for methods of efficient fine-tuning. Recently, an orthogonal fine-tuning paradigm was introduced that uses orthogonal matrices for adapting the weights of a pretrained model. In this paper, we introduce a new class of structured matrices, which unifies and generalizes structured classes from previous works. We examine properties of this class and build a structured orthogonal parametrization upon it. We then use this parametrization to modify the orthogonal fine-tuning framework, improving parameter and computational efficiency. We empirically validate our method on different domains, including adapting of text-to-image diffusion models and downstream task fine-tuning in language modeling. Additionally, we adapt our construction for orthogonal convolutions and conduct experiments with 1-Lipschitz neural networks.

## 1 Introduction

Orthogonal transforms have proven useful in different deep learning tasks. For example, they were shown to stabilize CNNs [Li et al., 2019, Singla and Feizi, 2021] or used in RNNs to combat the problem of exploding/vanishing gradients [Arjovsky et al., 2016]. Recent works OFT (Orthogonal Fine-Tuning) and BOFT (Butterfly Orthogonal Fine-Tuning) [Qiu et al., 2023, Liu et al., 2024b] use learnable orthogonal matrices for parameter-efficient fine-tuning of neural networks, which prevents training instabilities and overfitting that alternative methods like LoRA [Hu et al., 2022] suffer from.

Nevertheless, parametrization of orthogonal matrices is a challenging task, and the existing methods typically lack in either computational efficiency or expressiveness. Classical methods like Cayley parametrization and matrix exponential map cannot operate under low parameter budget, while Givens rotations and Householder reflections requires computing products of several matrices, which makes their use less efficient in deep learning tasks. Alternative approach in OFT method uses block-diagonal matrix structure in an attempt to be more computationally efficient and use less trainable parameters. Unfortunately, this simple structure can be too restrictive. Thus, arises the problem of constructing dense orthogonal matrix while still being parameter-efficient. While attempting to tackle this task, BOFT method uses a variation of butterfly matrices, parametrizing orthogonal matrices as a product of several matrices with different sparsity patterns, enforcing orthogonality on each of them. This parametrization is able to construct dense matrices while still being parameter-efficient. However it requires to compute a product of multiple matrices (typically up to 6) which can be computationally expensive. In this paper, we aim to overcome these issues and build dense orthogonal matrices in a more efficient way.

38th Conference on Neural Information Processing Systems (NeurIPS 2024).

We present a novel structured matrix class parametrized by an alternating product of block-diagonal matrices and several permutations. Multiplying by these matrices can be seen as a consecutive application of independent linear transforms within certain small groups and then shuffling the elements between them, hence the name Group-and-Shuffle matrices (or $\mathcal{GS}$-matrices for short). This class generalizes Monarch matrices [Dao et al., 2022] and with the right permutation choices, is able to form dense orthogonal matrices more effectively compared to approach proposed in BOFT, decreasing number of matrices in the product as well as the number of trainable parameters. We build efficient structured orthogonal parametrization with this class and use it to construct a new parameter-efficient fine-tuning method named GSOFT.

Our contributions:

- We introduce a new class of structured matrices, called $\mathcal{GS}$, that is more effective at forming dense matrices than block butterfly matrices from the BOFT method.
- Using $\mathcal{GS}$-matrices, we propose an efficient structured orthogonal parametrization, provide theoretical insights and study its performance in the orthogonal fine-tuning framework.
- We adapt our ideas for convolutional architectures, providing a framework to compress and speed-up orthogonal convolution layers.

## 2 Orthogonal Fine-tuning

Orthogonal Fine-tuning method (OFT) introduced in [Qiu et al., 2023] is a Parameter-Efficient Fine-Tuning (PEFT) method which fine-tunes pre-trained weight matrices through a learnable orthogonal block-diagonal matrix. Some of the properties that make orthogonal transforms desirable are preservation of pair-wise angles of neurons, spectral properties and hyperspherical energy. More precisely, OFT optimizes an orthogonal matrix $Q \in \mathbb{R}^{d \times d}$ for a pre-trained frozen weight matrix $W^0 \in \mathbb{R}^{d \times n}$ and modifies the multiplication $y = (W^0)^\top x$ to $y = (QW^0)^\top x$. Note that the identity matrix $I$ is orthogonal, which makes it a natural initialization for $Q$. OFT uses block-diagonal structure for $Q$, parameterizing it as

$$Q = \mathrm{diag}(Q_1, Q_2, \ldots, Q_r),$$

where $Q_i \in \mathbb{R}^{b \times b}$ are small orthogonal matrices and $br = d$. Orthogonality is enforced by Cayley parametrization, i.e.

$$Q_i = (I + K_i)(I - K_i)^{-1},$$

where $K_i$ are skew-symmetric: $K_i = -K_i^\top$. This ensures orthogonality of $Q_i$ and, hence, of $Q$.

Nevertheless, block-diagonal matrices can be too restrictive, as they divide neurons into $r$ independent groups based on their indices. This motivates the construction of dense parameter-efficient orthogonal matrices. To address this problem, the Orthogonal Butterfly method (BOFT) was introduced [Liu et al., 2024b]. BOFT uses block-butterfly structure to construct $Q$. Essentially, $Q$ is parameterized as a product of $m$ orthogonal sparse matrices:

$$Q = B_m B_{m-1} \ldots B_1.$$

Each matrix $B_i$ is a block-diagonal matrix up to a permutation of rows and columns, consisting of $r$ block matrices of sizes $b \times b$. Similarly to OFT, the orthogonality is enforced by the Cayley parametrization applied to each block. However, BOFT method has some areas for improvement as well. To construct a dense matrix, BOFT requires at least

$$m = 1 + \lceil \log_2(r) \rceil$$

matrices. For example, the authors of BOFT use $m = 5$ or 6 matrices in the BOFT method for fine-tuning of Stable Diffusion [Rombach et al., 2022]. Large amount of stacked matrices leads to significant time and memory overhead during training. There is also a room for improvement in terms of parameter-efficiency. To overcome these issues, we introduce a new class of structured matrices that we denote $\mathcal{GS}$ (group-and-shuffle) that generalizes Monarch matrices [Dao et al., 2022, Fu et al., 2024] and show how to use this class to construct parameter-efficient orthogonal parametrization. Similarly to BOFT, our approach uses block-diagonal matrices and permutations, but requires only

$$m = 1 + \lceil \log_b(r) \rceil$$

matrices of the same size to construct a dense matrix. See details in Section 5.2. The reduced requirements on $m$ allow us to use $m = 2$ in experiments to maximize computational efficiency, while still maintaining accurate results.

# 3 $\mathcal{GS}$-matrices

Our motivation within this work is to utilize orthogonal matrices of the form:

$$A = P_L(LPR)P_R \tag{1}$$

where matrices $L$ and $R$ are block-diagonal matrices with $r$ blocks of sizes $b \times b$ and $P_L, P, P_R$ are certain permutation matrices, e.g. $P_L = P^\top, P_R = I$ in the orthogonal fine-tuning setting and $P_R = P, P_L = I$ for convolutional architectures. Note that although the case $P_L = P^\top, P_R = I$ resembles Monarch matrices [Dao et al., 2022], they are unable to form such a structure, e.g., with equal-sized blocks in $L$ and $R$. The issue is that the Monarch class has a constraint that interconnects the number of blocks in matrix $L$ and the number of blocks in matrix $R$ (see Appendix C for details). Moreover, Monarch matrices have not been considered with orthogonality constraints.

To build matrices of the form (1), we first introduce a general class of $\mathcal{GS}$-matrices and study its properties. We then discuss orthogonal matrices from this class in Section 4.

## 3.1 Definition of $\mathcal{GS}$-matrices

**Definition 3.1.** *An $m \times n$ matrix $A$ is in $\mathcal{GS}(P_L, P, P_R)$ class with $k_L, k_R$ blocks and block sizes $b_L^1 \times b_L^2, b_R^1 \times b_R^2$ if*

$$A = P_L(LPR)P_R,$$

*where $L = \mathrm{diag}(L_1, L_2, \ldots, L_{k_L}), L_i \in \mathbb{R}^{b_L^1 \times b_L^2}$, $R = \mathrm{diag}(R_1, R_2, \ldots, R_{k_R}), R_i \in \mathbb{R}^{b_R^1 \times b_R^2}$, $P_L, P, P_R$ are permutation matrices and $b_L^2 \cdot k_L = b_R^1 \cdot k_R = s, b_L^1 \cdot k_L = m, b_R^2 \cdot k_R = n$.*

In practice, we fix $P_L, P, P_R$ depending on the application and only make matrices $L, R$ subject for change. $\mathcal{GS}$-matrices are hardware-efficient, as they are parametrized by two simple types of operations that can implemented efficiently: multiplications by block-diagonal matrices and permutations.

Let us also illustrate a forward pass $Ax \equiv LPRx$ for a matrix $A \in \mathcal{GS}(I, P, I)$ as a building block for the more general class with two additional permutations. The first operation $y = Rx$ consists of several fully-connected layers, applied individually to subgroups of $x$, see Figure 1. The next multiplication $LPy$ ensures that these groups interact with each other. Indeed, the permutation matrix $P$ shuffles the entries of $y$ into new subgroups. These subgroups are then again processed by a number of fully-connected layers using $L$. This motivates the naming for our class of matrices: *Group-and-Shuffle* or $\mathcal{GS}$ for short.

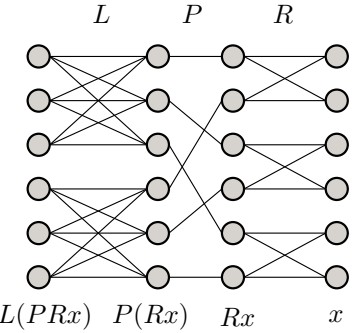

Figure 1: $\mathcal{GS}(I, P, I)$ matrices with $b_L^1 = b_L^2 = 3, b_R^1 = b_R^2 = 2$, $k_L = 2, k_R = 3$. Edges between nodes denote nonzero weights.

Another useful insight on these matrices is that the class $\mathcal{GS}(I, P, I)$ consists of block matrices with low-rank blocks. The permutation matrix $P$ is responsible for the formation of these blocks and defines their ranks (note that rank may vary from block to block). The result below formally describes our findings and is key to the projection operation that we describe afterwards.

**Proposition 1.** *Let $A$ be a matrix from $\mathcal{GS}(I, P, I)$ with a permutation matrix $P$ defined by the function $\sigma : \{0, \ldots, n-1\} \to \{0, \ldots, n-1\}$. Let $\{v_i^\top\}$ – be the rows of the blocks $R_1, \ldots, R_{k_R}$, $\{u_i\}$ – the columns of the blocks $L_1, \ldots, L_{k_L}$ in the consecutive order. Then the matrix $A$ can be written as a block matrix with $k_L \times k_R$ blocks using the following formula for each block $A_{k_1, k_2}$:*

$$A_{k_1, k_2} = \sum_{\substack{\lfloor \frac{\sigma(i)}{k_L} \rfloor = k_1 \\ \lfloor \frac{i}{k_R} \rfloor = k_2}} u_{\sigma(i)} v_i^\top.$$

*Note that we use zero-indexing for this proposition for simplicity of formulas.*

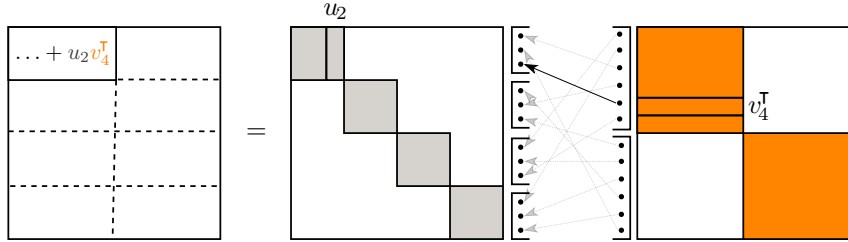

Figure 2: Illustration of Proposition 1 that provides block low-rank interpretation of $\mathcal{GS}(I, P, I)$ matrices. The matrix $R$ contains 2 blocks and matrix $L$ contains 4 blocks.

Let us illustrate this proposition in Figure 2. We consider $\mathcal{GS}(I, P, I)$ with $k_L = 4$ and $k_R = 2$ blocks in $L$ and $R$ and with the block sizes $3 \times 3$ and $6 \times 6$ respectively. Let us consider the leading block $A_{00}$ of the size $3 \times 6$. According to Proposition 1, $A_{00} = u_0 v_2^\top + u_2 v_4^\top$. Indeed, let us take a closer look, e.g., at the term $u_2 v_4^\top$. In the permutation matrix $P$, we have a nonzero element in the position $(2, 4)$ as $i = 4$ and $\sigma(4) = 2$. Therefore, we select the third column $u_2$ in $L_1$ and the fifth row $v_4^\top$ in $R_1$. This leads to adding a rank-one term $u_2 v_4^\top$ to $A_{00}$ as we see in the formula above.

Another direct corollary from Proposition 1 is a projection operation $\pi \colon \mathbb{R}^{m \times n} \to \mathcal{GS}(P_L, P, P_R)$ that satisfies:
$$\pi(A) \in \underset{B \in \mathcal{GS}(P_L, P, P_R)}{\arg\min} \|A - B\|_F,$$
where $\| \cdot \|_F$ is the Frobenius norm. Thanks to the block low-rank representation of matrices from $\mathcal{GS}(P_L, P, P_R)$, the projection $\pi$ is simply constructed using SVD truncations of the blocks $(P_L^\top A P_R^\top)_{k_1, k_2}$ and is summarized in Algorithm 1.

---

**Algorithm 1** Projection $\pi(\cdot)$ of $A$ onto $\mathcal{GS}(P_L, P, P_R)$

---

**Input:** $A, P_L, P, P_R$
**Return:** $L, R$
**for** $k_1 = 1 \ldots k_L$ **do**
    **for** $k_2 = 1 \ldots k_R$ **do**
        Compute SVD of $(P_L^T A P_R^T)_{k_1, k_2} = U \Sigma V^\top$;
        Set $r = r_{k_1, k_2}$ – rank of block determined by $P$;
        Take $U_r = U[:r, :], \Sigma_r = \Sigma[:r, :r], V_r = V[:r, :]$;
        Pack columns of $U_r \Sigma_r^{1/2}$ into $L_{k_1}$ and rows of $\Sigma_r^{1/2} V_r$ into $R_{k_2}$ according to $P$;
    **end for**
**end for**

---

## 4 Orthogonal $\mathcal{GS}(P_L, P, P_R)$ matrices

In this section, we study the orthogonality constraint for the $\mathcal{GS}(P_L, P, P_R)$ to obtain structured orthogonal representation. This is one of the main contributions of our paper and we utilize this class in all the numerical experiments. Since we are interested only in square orthogonal matrices, we additionally assume that $m = n$ and $b_L^1 = b_L^2 = b_L$; $b_R^1 = b_R^2 = b_R$. Similarly to parametrizations in OFT and BOFT, a natural way to enforce orthogonality of $\mathcal{GS}(P_L, P, P_R)$-matrices is to enforce orthogonality of each block of $L$ and $R$. This indeed leads an orthogonal matrix since permutation matrices are also orthogonal as well as a product of orthogonal matrices. However, it is not immediately obvious that there exist no orthogonal matrices from $\mathcal{GS}(P_L, P, P_R)$ that cannot be represented this way. Surprisingly, we find that such a way to enforce orthogonality is indeed sufficient for covering of all orthogonal matrices from $\mathcal{GS}(P_L, P, P_R)$.

**Theorem 1.** *Let $A$ be any orthogonal matrix from $\mathcal{GS}(P_L, P, P_R)$. Then, $A$ admits $P_L(LPR)P_R$ representation with the matrices $L, R$ consisting of orthogonal blocks.*

*Proof.* Matrices $P_L, P_R$ are orthogonal as they are permutation matrices. It means that it is sufficient to prove theorem in the case when $A$ is from $\mathcal{GS}(I, P, I)$, which means that we can use low block-

rank structure interpretation from Proposition 1. Consider a skeleton decomposition of the blocks $A_{ij} = U_{ij}V_{ij}^\top$, $U_{ij} \in \mathbb{R}^{b_L \times r_{ij}}$, $V_{ij} \in \mathbb{R}^{b_R \times r_{ij}}$ such that $U_{ij}^\top U_{ij} = I_{r_{ij}}$ (this can be ensured, e.g., using the QR decomposition). Then

$$
A = \begin{pmatrix} U_{1,1}V_{1,1}^\top & \cdots & U_{1,k_L}V_{1,k_L}^\top \\ \vdots & \ddots & \vdots \\ U_{k_L,1}V_{k_L,1}^\top & \cdots & U_{k_L,k_R}V_{k_L,k_R}^\top \end{pmatrix}.
$$

Take the $j$-th block-column of $A$. Since $A$ is an orthogonal matrix, we get:

$$
\begin{pmatrix} V_{1,j}U_{1,j}^\top & \cdots & V_{k_L,j}U_{k_L,j}^\top \end{pmatrix} \begin{pmatrix} U_{1,j}V_{1,j}^\top \\ \vdots \\ U_{k_L,j}V_{k_L,j}^\top \end{pmatrix} = I_{b_R}
$$

Multiplying matrices in the l.h.s. we get $V_{1,j}U_{1,j}^\top U_{1,j}V_{1,j}^\top + \cdots + V_{k_L,j}U_{k_L,j}^\top U_{k_L,j}V_{k_L,j}^\top = I_{b_R}$. Since $U_{ij}^\top U_{ij} = I_{r_{ij}}$ we conclude $V_{1,j}V_{1,j}^\top + \cdots + V_{k_L,j}V_{k_L,j}^\top = I_{b_R}$. This implies that $\begin{pmatrix} V_{1,j} & \cdots & V_{k_L,j} \end{pmatrix}$ is an orthogonal matrix. Note that if we now parameterize $A = LPR$ with the matrices $V_{ij}$ packed into $R$ and $U_{ij}$ packed into $L$, then $\begin{pmatrix} V_{1,j} & \cdots & V_{k_L,j} \end{pmatrix}$ is exactly the $j$-th block matrix in $R$ up to permutation of rows. Therefore, every block in $R$ is an orthogonal matrix. Since we now proved that $V_{ij}^\top V_{ij} = I$, we can use same the derivation for the rows of $A$ and conclude that blocks of $L$ are also orthogonal. $\qquad\square$

# 5 $\mathcal{GS}(P_{m+1}, \ldots, P_1)$ matrices

In this section we describe an the extension of $\mathcal{GS}$-matrices that uses more than two block-diagonal matrices and show that with the right permutations choices $\mathcal{GS}$-matrices are more effective than block butterfly matrices in forming dense matrices. Here by dense matrices we imply matrices that do not contain zero entries at all.

**Definition 5.1.** *A is said to be in $\mathcal{GS}(P_{m+1}, \ldots, P_1)$ if*

$$
A = P_{m+1} \prod_{i=m}^{1} (B_i P_i),
$$

*where each matrix $B_i$ is a block-diagonal matrix with $k_i$ blocks of size $b_i^1 \times b_i^2$, matrices $P_i$ are permutation matrices and $b_i^1 \cdot k_i = b_{i+1}^2 \cdot k_{i+1}$.*

**Remark 1.** *Similarly to the case $m = 2$ described in Section 3, we may use orthogonal blocks in $B_i$, $i = 1, \ldots, m+1$ to obtain orthogonal matrices. However, it is not clear if an analog to Theorem 1 is correct in this case as well.*

**Remark 2.** *For each of the classes of Block Butterfly matrices [Chen et al., 2022], Monarch matrices [Dao et al., 2022] and order-p Monarch matrices [Fu et al., 2024], there exist permutation matrices $P_{m+1}, \ldots, P_1$ such that $\mathcal{GS}(P_{m+1}, \ldots P_1)$ coincides with a respective class. Indeed, Monarch matrices have the form of alternating block-diagonal matrices and permutations with some specific size constraints and sparse matrices in the product of Block Butterfly matrices can be easily transformed to block-diagonal matrices with permutations of rows and columns.*

## 5.1 Choosing permutation matrices

We suggest using the following matrices with $k = k_i$ for $P_i$. Note that this is efficient for forming dense matrices as follows from the proof of Theorem 2. This is by contrast to the permutations used in [Fu et al., 2024] that are restricted to particular matrix sizes.

**Definition 5.2** ([Dao et al., 2022]). *Let $P_{(k,n)}$ be a permutation matrix given by permutation $\sigma$ on $\{0, 1, \ldots, n-1\}$:*

$$
\sigma(i) = (i \bmod k) \cdot \frac{n}{k} + \left\lfloor \frac{i}{k} \right\rfloor.
$$

Applying this permutation to a vector can be viewed as reshaping an input of size $n$ into an $k \times \frac{n}{k}$ matrix in a row-major order, transposing it, and then vectorizing the result back into a vector (again in row-major column). We provide several examples of such permutations in Figure 3.

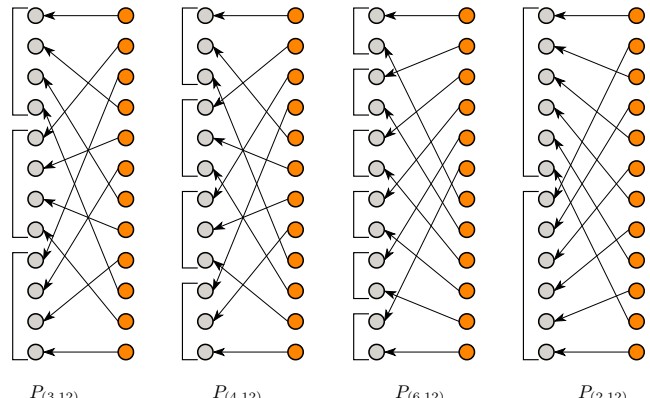

$$P_{(3,12)} \qquad P_{(4,12)} \qquad P_{(6,12)} \qquad P_{(2,12)}$$

Figure 3: Illustraion of $P_{(k,12)}$ permutations for $k \in \{3, 4, 6, 2\}$.

## 5.2 Comparison to block butterfly matrices and BOFT

Block Butterfly matrices were introduced in [Chen et al., 2022] and are used to construct orthogonal matrices in the BOFT method. Block Butterfly matrix class is a special case of higher-order $\mathcal{GS}$-matrices with $k_i = r$ and $b_i^1 = b_i^2 = b = 2s$ and certain permutation choices. However, we argue that the choice of these permutations are sub-optimal for construction of dense matrix and using permutations from Definition 5.2 is more effective. When using block-diagonal matrices with $r$ blocks, block butterfly matrices need $1 + \lceil \log_2(r) \rceil$ matrices to construct a dense matrix. For $\mathcal{GS}$-matrices we have the following result.

**Theorem 2.** *Let $k_i = r, b_i^1 = b_i^2 = b$. Then using $m = 1 + \lceil \log_b(r) \rceil$ is sufficient for the class $\mathcal{GS}(P_L, P_{(r,br)}, \ldots, P_{(r,br)}, P_R)$ to form a dense matrix for any $P_L, P_R$. Moreover, the choice of $P_2 = \cdots = P_m = P_{(r,br)}$ is optimal in the sense that all matrices from $\mathcal{GS}(P_{m+1}, \ldots, P_1)$ contain zero blocks for any integer $m < 1 + \lceil \log_b(r) \rceil$ and any permutations $P_1, \ldots, P_{m+1}$.*

*Proof.* See Appendix D. □

For example, let us consider a case of constructing a dense orthogonal matrix of the size $1024 \times 1024$. Suppose also that we use block matrices with block size 32. Constructing a dense matrix with Block Butterfly matrices requires $1 + \log_2(32) = 6$ butterfly matrices, which leads to $6 \times 32^3$ parameters in the representation. $\mathcal{GS}(P_L, P, P_R)$ matrices with $P = P_{(32,1024)}$ only need two matrices to construct a dense matrix yielding $2 \times 32^3$ parameters. The $\mathcal{GS}(P_L, P, P_R)$ parametrization is also naturally more computationally efficient as fewer number of multiplications is both faster and requires less cached memory for activations.

## 6 Applications

### 6.1 Orthogonal fine-tuning with $\mathcal{GS}(P_L, P, P_R)$ (GSOFT)

We utilize the pipeline of OFT and BOFT methods with the exception of parametrizing $Q$ with orthogonal permuted $\mathcal{GS}(P_L, P, P_R)$ matrices. In particular, for parametrization of $Q \in \mathbb{R}^{d \times d}$, we utilize the $\mathcal{GS}(P^\top, P, I)$ class, i.e. $Q = P^\top L P R$, where $L = \mathrm{diag}(L_1, \ldots L_r)$, $L_i \in \mathbb{R}^{b \times b}$, $R = \mathrm{diag}(R_1, \ldots, R_r)$, $R_i \in \mathbb{R}^{b \times b}$. For consistency, we use the same notation for the number of blocks and block sizes as in BOFT and OFT methods. We use $P_{(r,br)}$ as a permutation matrix $P$. To enforce orthogonality, we parameterize each block in matrices $L, R$ with the Cayley parametrization. We initialize $Q$ as an identity matrix by initializing each block to be an identity matrix. Additional

techniques like magnitude scaling and multiplicative dropout that are used in OFT and BOFT can be utilized the same way in our method, though we only use scaling in our experiments. Note that likewise in OFT, BOFT weights of the matrix $Q$ can be merged with the pretrained weight $W$ producing no inference overhead.

## 6.2 Two-sided orthogonal fine-tuning (Double GSOFT)

Consider SVD decomposition of a matrix $W^0 = U\Sigma V^\top$. Applying orthogonal fine-tuning, we get $W' = (QU)\Sigma V^\top$, which is an SVD decomposition for the adapted weight $W'$. This shows that we can only change left singular vectors $U$ with the standard orthogonal fine-tuning paradigm. At the same time, the LoRA method modifies both matrices $U$ and $V$. Moreover, recent papers [Meng et al., 2024, Li et al., 2023] show that initializing matrices $A, B$ with singular vectors can additionally boost performance of LoRA. This motivates an extension of orthogonal fine-tuning method, that can adapt both matrices $U$ and $V$. We introduce a simple approach that multiplies pre-trained weight matrices from both sides, rather than one. This method modifies forward pass from z $= (W^0)^\top x$ to

$$z = (Q_U W^0 Q_V)^\top x$$

Where $Q_U$ and $Q_V$ are parametrized as orthogonal $\mathcal{GS}$-matrices. In cases where BOFT utilizes 5-6 matrices, we can leverage the fact that our method uses only 2 and adapt both sides while still using less matrices and trainable parameters than BOFT.

## 6.3 $\mathcal{GS}$ Orthogonal Convolutions

Recall, that due to linearity of a multichannel convolution operation, we can express the convolution of tensor $X \in \mathbb{R}^{c_{in} \times h \times w}$ with a kernel $L \in \mathbb{R}^{c_{out} \times c_{in} \times k \times k}$ $L \star X$ in terms of matrix multiplication [Singla and Feizi, 2021]:

$$Y = L \star X \quad \Leftrightarrow \quad vec(Y) = \begin{bmatrix} L_{0,0} & \cdots & L_{0,c_{in}-1} \\ \vdots & \ddots & \vdots \\ L_{c_{out}-1,0} & \cdots & L_{c_{out}-1,c_{in}-1} \end{bmatrix} vec(X), \tag{2}$$

where $L_{i,j}$ is doubly Toeplitz matrix, corresponding to convolution between $i$-th and $j$-th channels and $vec(X)$ is a vectorization of tensor into a vector in a row-major order. Thus, the convolution is essentially a block matrix, where each block represents a standard convolution operation. Using this block interpretation (2), we may apply the concept of $\mathcal{GS}$ matrices to convolutional layers as well. Considering each convolution between channels as an element of our block matrix, we can set some of these blocks to zero, obtaining some additional structure. Thus, we can construct block matrix which has block-diagonal structure, corresponding to grouped convolution (further, in all equations we will denote it as GrConv). Then, defining ChShuffle as a permutation of channels, like in [Zhang et al., 2018], we obtain structure, which is similar to GSOFT, defined in Section 6:

$$Y = \texttt{GrConv}_2(\texttt{ChShuffle}_2(\texttt{GrConv}_1(\texttt{ChShuffle}_1(X)))). \tag{3}$$

The proposed $\mathcal{GS}$ convolutional layer shuffles information between each pair of input channels and requires less parameters and FLOPs during computations. In this example we can also choose permutations of channels and change kernel size. This convolutional layer can be treated as $\mathcal{GS}(P_{m+1}, \ldots, P_1)$ matrix in vectorized view, that is why choosing permutations between convolutional layers is also very important for information transition properties. In Appendix F we explain the choice of ChShuffle operation.

We can use the proposed layer to construct orthogonal convolutions (transformations with an orthogonal Jacobian matrix) similarly to skew orthogonal convolution (SOC) architecture, that uses Taylor expansion of a matrix exponential. One major downside of methods such as SOC and BCOP [Li et al., 2019] is that they require more time than basic convolution operation. For instance, in the SOC method, one layer requires multiple applications of convolution (6 convolutions per layer). In our framework, we propose a parametrization of a convolutional layer, in which imposing an orthogonality to convolutions has fewer number of FLOPs and parameters thanks to the usage of grouped convolutions.

Let us discuss in more details how SOC works and the way we modify it. In SOC, a convolutional filter is parametrized in the following way:

$$L = M - \texttt{ConvTranspose}(M),$$

where $M \in \mathbb{R}^{c_{in} \times c_{out} \times r \times s}$ is an arbitrary kernel and the $\texttt{ConvTranspose}$ is the following operation:

$$\texttt{ConvTranspose}(M)_{i,j,k,l} = M_{j,i,r-k-1,s-l-1}$$

This parametrization of filter $L$ makes the matrix from Equation 2 skew-symmetric. As matrix exponential of skew-symmetric matrix is an orthogonal matrix, in SOC the authors define convolution exponential operation, which is equivalent to matrix exponential in matrix-vector notation:

**Definition 6.1.** *[Singla and Feizi, 2021] Let $X \in \mathbb{R}^{c \times h \times w}$ be an input tensor and $L \in \mathbb{R}^{c \times c \times k \times k}$ be a convolution kernel. Then, define convolution exponential $L \star_e X$ as follows:*

$$L \star_e X = X + \frac{L \star X}{1!} + \frac{L \star^2 X}{2!} + \dots$$

*where $L \star^i X$ is a convolution with kernel $L$ applied $i$ times consequently.*

As mentioned above, with proper initialization we get a convolutional layer with orthogonal Jacobian matrix. Using the parametrization of convolution layer from the Equation 3 and substituting there two grouped convolution exponentials (e.g. in our parametrization we have the same convolution exponential, but we have grouped convolution instead of basic one) with the parameterized kernel:

$$Y = \texttt{GrExpConv}_2(\texttt{ChShuffle}_2(\texttt{GrExpConv}_1(\texttt{ChShuffle}_1(X))))$$

In our experiments we tried different layer architectures and we found that making kernel size of the second convolution equal to 1 speeds up our convolutional layer, maintaining quality metrics. Thus, if convolutional layer consists of two grouped convolutional exponentials, the second convolutional exponential has $kernel\_size = 1 \times 1$

## 7 Experiments

All the experiments below were conducted on NVIDIA V100-SXM2-32Gb GPU. We ran all the experiments within $\sim$2000 GPU hours.

Source code is available at: https://github.com/Skonor/group_and_shuffle

### 7.1 Natural language understanding

We report result on the GLUE [Wang et al., 2018] benchmark with RoBERTa-base [Liu et al., 2019] model. Benchmark includes several classification tasks that evaluate general language understanding. We follow training settings of [Liu et al., 2024b, Zhang et al., 2023]. We apply adapters for all linear layers and only tune learning rate for all methods. Table 1 reports best results on the evaluation set from the whole training. LoRA, OFT and BOFT are implemented with PEFT library [Mangrulkar et al., 2022]. GSOFT method outperforms OFT, BOFT and also have a slight edge over LoRA. Note that even though skew-symmetric $K$ theoretically matrix only requires approximately half the parameters of a full matrix, in practice it is parametrized as $K = A - A^T$ for the ease of computations. However, after fine-tuning, one can only save upper-triangular part of $K$. Doing this, orthogonal fine-tuning methods become approximately 2 times more efficient in terms of memory savings.

### 7.2 Subject-driven generation

Subject-driven generation [Ruiz et al., 2023, Gal et al., 2022] is an important and challenging task in the field of generative modelling. Given several photos of a particular concept, we want to introduce it to the diffusion model so that we can generate this particular object in different scenes described by textual prompts. The main way to do this is to fine-tune the model. However, the large number of fine-tuning parameters together with the lack of training images make the model prone to overfitting, i.e. the model reconstructs the concept almost perfectly, but starts to ignore the textual prompt during generation. To solve this problem and stabilize the fine-tuning process, different lightweight parameterizations [Qiu et al., 2023, Liu et al., 2024b, Hu et al., 2022, Tewel et al., 2023, Han et al.,

Table 1: Results on GLUE benchmark with RoBERTa-base model. We report Pearson correlation for STS-B, Matthew's correlation for CoLA and accuracy for other tasks. # Params denotes number of trainable parameters

| Method | # Params | MNLI | SST-2 | CoLA | QQP | QNLI | RTE | MRPC | STS-B | ALL |
|--------|----------|------|-------|------|-----|------|-----|------|-------|-----|
| FT | 125M | 87.62 | 94.38 | 61.97 | **91.5** | 93.06 | 80.14 | 88.97 | **90.91** | 86.07 |
| LoRA$_{r=8}$ | 1.33M | **87.82** | **95.07** | 64.02 | 90.97 | 92.81 | **81.95** | 88.73 | 90.84 | 86.53 |
| OFT$_{b=16}$ | 1.41M | 87.21 | **95.07** | 64.37 | 90.6 | 92.48 | 79.78 | 89.95 | 90.71 | 86.27 |
| BOFT$_{b=8}^{m=2}$ | 1.42M | 87.14 | 94.38 | 62.57 | 90.48 | 92.39 | 80.14 | 88.97 | 90.67 | 85.84 |
| GSOFT$_{b=8}$ | 1.42M | 87.16 | **95.06** | **65.3** | 90.46 | 92.46 | **81.95** | **90.2** | 90.76 | **86.67** |

Table 2: Results on subject-driven generation. # Params denotes the number of training parameters in each parametrization. Training time is computed for 3000 iterations on a single GPU V100 in hours.

| Model | Full | LoRA | | | BOFT | | | GSOFT (Ours) | | | Double GSOFT (Ours) | | |
|-------|------|------|------|------|------|------|------|------|------|------|------|------|------|
| | | rank | | | $r, m$ | | | $r$ | | | $r$ | | |
| | | 4 | 32 | 128 | 32, 4 | 32, 6 | 16, 5 | 32 | 16 | 8 | 64 | 32 | 16 |
| # Params | 99.9M | 0.8M | 6.6M | 26.6M | 13.6M | 20.4M | 33.8M | 6.8M | 13.6M | 27.1M | 6.5M | 13.0M | 25.9M |
| Training time | 1.3 | 1.3 | 1.3 | 1.3 | 2.0 | 2.2 | 2.3 | 1.5 | 1.6 | 1.8 | 1.7 | 2.0 | 1.8 |
| CLIP-I ↑ | 0.805 | 0.805 | **0.819** | 0.813 | 0.803 | 0.796 | 0.789 | 0.805 | 0.803 | 0.783 | **0.815** | 0.802 | 0.783 |
| CLIP-T ↑ | 0.212 | 0.246 | 0.236 | 0.223 | 0.244 | 0.234 | 0.223 | **0.256** | 0.245 | 0.227 | **0.256** | 0.242 | 0.225 |

2023] and regularization techniques [Ruiz et al., 2023, Kumari et al., 2023] are widely used in this task. Therefore, we chose this setting to evaluate the effectiveness of the proposed orthogonal parameterization compared to other approaches.

We use StableDiffusion [Rombach et al., 2022] and the Dreambooth [Ruiz et al., 2023] dataset for all our experiments. The following parameterizations were considered as baselines in this task: full (q, k, v and out.0 layers in all cross- and self- attentions of the UNet are trained), LoRA [Hu et al., 2022] and BOFT [Liu et al., 2024b] applied to the same layer. We use our GSOFT parameterization and a two-sided orthogonal GSOFT (Double GSOFT) applied to the same layers as baselines. For a more comprehensive comparison, we consider different hyperparameters for the models, adjusting the total number of optimized parameters. More training and evaluation details can be found in Appendix E.

CLIP image similarity, CLIP text similarity and visual comparison for this task are presented in Table 2 and Figure 4. As the results show, GSOFT and DoubleGSOFT are less prone to overfitting compared to the baselines. They show better alignment with text prompts while maintaining a high level of concept fidelity. Furthermore, both methods with optimal hyperparameters are more efficient than BOFT and comparable to LoRA and full parameterization in terms of training time. See Appendix E for more visual and quantitative comparison.

### 7.3 $\mathcal{GS}$ Orthogonal Convolutions

Following [Singla and Feizi, 2021], we train LipConvnet-n on CIFAR-100 dataset. LipConvnet-n is 1-Lipschitz neural network, i.e. neural network with Lipschitz constant equal to 1, his property provides certified adversarial robustness. LipConvnet uses orthogonal convolutions and gradient preserving activations in order to maintain 1-Lipschitz property.

LipConvnet-n architecture consists of 5 equal blocks, each having $\frac{n}{5}$ skew orthogonal convolutions, where the last convolution at each level downsamples image size. We replace the skew orthogonal convolution layer with the structured version using $\mathcal{GS}$orthogonal convolutions and test it in the setting of [Singla and Feizi, 2021], using the same hyperparameters (learning rate, batch size and scheduler stable during testing). In layers where we have two GrExpConv, the second convolution has kernel size equal to 1.

We also use a modified activation function (MaxMinPermuted instead of MaxMin), which uses different pairing of channels. This makes activations aligned with the ChShuffle operation and grouped convolutions. The choice of permutation for ChShuffle also slightly differs from permutations defind in Definition 5.2 because of the interplay between activations and convolutional layers. We provide definitions and intuition regarding activations and permutations for ChShuffle in Appenix F.

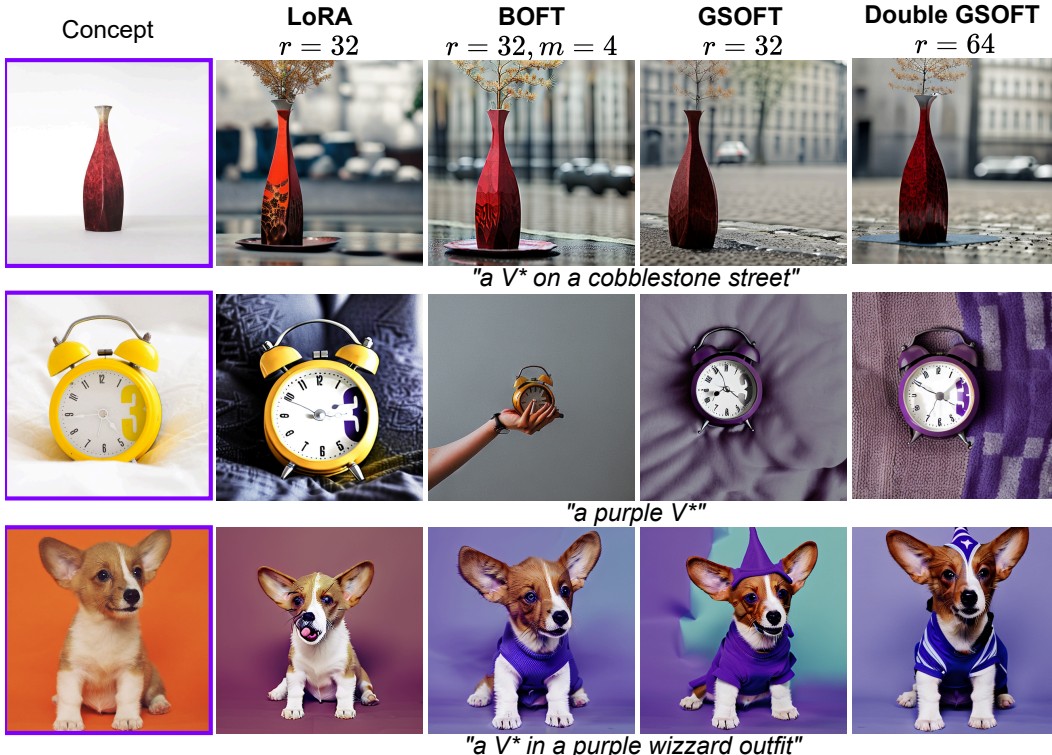

|  | **LoRA** $r = 32$ | **BOFT** $r = 32, m = 4$ | **GSOFT** $r = 32$ | **Double GSOFT** $r = 64$ |

Concept

*"a V* on a cobblestone street"*

*"a purple V*"*

*"a V* in a purple wizzard outfit"*

Figure 4: Subject-driven generation visual results on 3000 training iterations.

Table 3: Results of training LipConvnet-15 architecture on CIFAR-100. $(a, b)$ in "Groups" column denotes number of groups in two grouped exponential convolutions (with kernel sizes 3 and 1). $(a, -)$ corresponds to only one $\mathcal{GS}$ orthogonal convolutional layer. Before each grouped layer with $k$ groups use a `ChShuffle` operator.

| Conv. Layer | # Params | Groups | Speedup | Activation | Accuracy | Robust Accuracy |
|---|---|---|---|---|---|---|
| SOC | 24.1M | - | 1 | MaxMin | 43.15% | 29.18% |
| GS-SOC | **6.81M** | (4, -) | **1.64** | MaxMinPermuted | **43.48%** | **29.26%** |
| GS-SOC | **8.91M** | (4, 1) | 1.21 | MaxMinPermuted | **43.42%** | **29.56%** |
| GS-SOC | **7.86M** | (4, 2) | 1.22 | MaxMinPermuted | 42.86% | 28.98% |
| GS-SOC | **7.3M** | (4, 4) | 1.23 | MaxMinPermuted | 42.75% | 28.7% |

## 8 Concluding remarks

In this paper, we introduce a new class of structured matrices, called $\mathcal{GS}$-matrices, build a structured orthogonal parametrization with them and use them in several domains within deep learning applications. However, we hope that our orthogonal parametrization can be adapted to different settings in future (including tasks outside of deep learning), as it makes orthogonal parametrizations less of a computational burden. $\mathcal{GS}$-matrices without orthogonality constraints is another promising direction.

## 9 Limitations

Although our method for orthogonal fine-tuning is faster than BOFT, it is still slower than LoRA during training. Additionally, since our parametrization provides a trade-off between expressivity and parameter-efficiency, it might be unable to represent some particular orthogonal matrices, which might be required in other settings apart from parameter-efficient fine-tuning.

## 10 Acknowledgments

The article was prepared within the framework of the HSE University Basic Research Program. This research was supported in part through computational resources of HPC facilities at HSE University [Kostenetskiy et al., 2021].

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

# A  Related work

**Parameter-Efficient Fine-Tuning (PEFT)** With the growth of model sizes, end-to-end training became unavailable for those who want to adapt powerful architectures for specific tasks, as even full fine-tuning became too expensive. This problem sparked research in the direction of parameter-efficient fine-tuning methods, including methods that focus on prompt tuning [Lester et al., 2021, Li and Liang, 2021] and adapter tuning (e.g. [Houlsby et al., 2019, Karimi Mahabadi et al., 2021]), which include LoRA [Hu et al., 2022] and its variations [Meng et al., 2024, Zhang et al., 2023, Liu et al., 2024a, Dettmers et al., 2024, Li et al., 2023], that inject learnable low-rank matrices as an additive injection to the weights of pretrained models. OFT [Qiu et al., 2023], BOFT [Liu et al., 2024b] and our method use similar approach to LoRA, but learn multiplicative injection rather than an additive one.

**Structured sparsity** Structured sparsity is an approach that replaces dense weight layers with different structured ones, such as matrix factorizations or tensor decompositions in order to compress or speed-up models [Dao et al., 2022, Chen et al., 2022, Novikov et al., 2015, Lebedev et al., 2015]. Some of these techniques were also adapted to PEFT methods in works like [Karimi Mahabadi et al., 2021, Edalati et al., 2022, Yang et al., 2024] or BOFT [Liu et al., 2024b] method, that utilizes a variation of butterfly matrices as a parametrization for parameter-efficient orthogonal matrices, imposing orthogonality to each butterfly factor. See details in Section 2. Monarch matrices [Dao et al., 2022, Fu et al., 2024] are most relevant to our work as our proposed matrix class is their generalization that utilizes similar structure.

**Subject-driven generation** The emergence of large text-to-image models [Ramesh et al., 2022, 2021, Saharia et al., 2022, Rombach et al., 2022] has propelled the advancement of personalized generation techniques in the research field. Customizing a text-to-image model to generate specific concepts based on multiple input images presents a key challenge. Various methods [Ruiz et al., 2023, Gal et al., 2022, Kumari et al., 2023, Han et al., 2023, Qiu et al., 2023, Zhou et al., 2023, Wei et al., 2023, Tewel et al., 2023] have been proposed to address this challenge, requiring either extensive fine-tuning of the model as a whole [Ruiz et al., 2023] or specific parts [Kumari et al., 2023] to accurately reconstruct concept-related training images. While this facilitates precise learning of the input concept, it also raises concerns regarding overfitting, potentially limiting the model's flexibility in generating diverse outputs in response to different textual prompts. Efforts to mitigate overfitting and reduce computational burden have led to the development of lightweight parameterization techniques [Qiu et al., 2023, Liu et al., 2024b, Hu et al., 2022, Tewel et al., 2023, Han et al., 2023] such as those proposed among others. These methods aim to preserve editing capabilities while sacrificing some degree of concept fidelity. The primary objective is to identify parameterization strategies that enable high-quality concept learning without compromising the model's ability to edit and generate variations of the concept. Our investigation indicates that the orthogonal parameterization approach we propose represents a significant step towards achieving this goal.

**Orthogonal transforms** In papers [Vorontsov et al., 2017, Hyland and Rätsch, 2017, Helfrich et al., 2018] authors work in the setting of dense matrices and try to parameterize essentially the whole manifold of orthogonal matrices. For $n \times n$ matrices it requires computing inverses or exponential maps of $n \times n$ matrices at every optimization step. This takes $\mathcal{O}(n^3)$ time, which can be computationally challenging for larger architectures. Moreover, such a parametrization utilizes $\mathcal{O}(n^2)$ trainable parameters, which makes it inapplicable for PEFT (see the OFT paper [Qiu et al., 2023] for more details). Our proposed method is different in a sense that it provides a trade-off between expressivity (describing only a subset of the orthogonal manifold) and efficiency.

In [Li et al., 2019, Singla et al., 2021] authors discuss main issues of bounding of Lipschitz constant of neural networks and provide Gradient-Norm-Preserving (GNP) architecture in order to avoid vanishing of gradients while bounding Lipschitz constant. The authors propose a specific convolutional layer (Block Convolutional Orthogonal Parametrization) which Jacobian is orthogonal, also providing orthogonal activations with Lipschitz constant equal to 1. These constraints guarantee that the norm of the gradient will not change through backward pass. In other works [Singla and Feizi, 2021, Singla et al., 2021] authors provide a modification of the orthogonal convolutions (Skew Orthogonal Convolution) in terms of hardware-efficiency. Authors provide neural network architecture where each layer is 1-Lipschitz and make a comparison between these two convolutional layers. The work [Li et al., 2019] was outperformed by the SOC method that we utilize (there is a comparison in the SOC method [Singla and Feizi, 2021]). In [Xu et al., 2022], the authors utilize periodicity of

convolution and their padding is not equivalent to a widely-used zero-padded convolution. Survey [Prach et al., 2024] suggests that [Xu et al., 2022] and other 1-Lipschitz architectures yield worse robustness metrics than SOC, which we use as a baseline in our paper.

# B  Proof of Prop. 1

*Proof.* Let $R' = PR$. $R'$ can be viewed as a block matrix with $k_L \times k_R$ blocks of sizes $b_2^L \times b_2^R$. $L$ can be viewed as a block matrix with $k_L \times k_L$ blocks from which only diagonal are non-zero. The $A$ can be written in the following form:

$$
\begin{pmatrix} A_{0,0} & \cdots & A_{0,k_R-1} \\ \vdots & \ddots & \vdots \\ A_{k_L-1,0} & \cdots & A_{k_L-1,k_R-1} \end{pmatrix} = \begin{pmatrix} L_0 & \cdots & 0 \\ \vdots & \ddots & \vdots \\ 0 & \cdots & L_{k_L-1} \end{pmatrix} \begin{pmatrix} R'_{0,0} & \cdots & R'_{0,k_R-1} \\ \vdots & \ddots & \vdots \\ R'_{k_L-1,0} & \cdots & R'_{k_L-1,k_R-1} \end{pmatrix}.
$$

Using block matrix product formulas, we get:

$$
A_{k_1,k_2} = L_{k_1} R'_{k_1,k_2}.
$$

We can now rewrite $L_{k_1} R'_{k_1,k_2}$ product in terms of their columns and rows:

$$
L_{k_1} R'_{k_1,k_2} = \begin{pmatrix} l_1 \ldots l_{b_L^2} \end{pmatrix} \cdot \begin{pmatrix} r_1^\top \\ \vdots \\ r_{b_L^2}^\top \end{pmatrix} = \sum_t l_t r_t^\top. \tag{4}
$$

Columns of $L_{k_1}$ are just vectors $u_j$ such that $\lfloor \frac{j}{k_L} \rfloor = k_1$. Let us examine the rows of $R'_{k_1,k_2}$. Since $R'$ is a matrix formed by permuting the rows of block-diagonal matrix $R$, $R'_{k_1,k_2}$ can only contain rows that were in the $R_{k_2}$ before permutation. Formally, this means that $R'_{k_1,k_2}$ can only contain vector-rows $v_i^T$ such that $\lfloor \frac{i}{k_R} \rfloor = k_2$. Additionally, rows after permutation should get into the $k_1$-block row. That implies $\lfloor \frac{\sigma(i)}{k_L} \rfloor = k_1$. Other rows of $R'_{k_1,k_2}$ are zero-rows. Notice that in (4) non-zero rows $r_t^\top$ represented by $v_i^\top$ will match exactly with columns $u_{\sigma(i)}$ that represent $l_t$. Keeping only non-zero terms in $\sum_t l_t r_t^\top$ gets us to the desired conclusion. □

# C  Comparison of Monarch matrices and $\mathcal{GS}$-matrices

$\mathcal{GS}(P_L, P, P_r)$ class is inspired by Monarch matrices [Dao et al., 2022] and their primary goal is to introduce additional flexibility in the block structure of matrices $L$ and $R$. Generalized Monarch matrices are parameterized as $P_1 L P_2 R$, where $L$ and $R$ are block-diagonal matrices and $P_1$ and $P_2$ are certain permutations defined in Definition 5.2. This resembles $\mathcal{GS}(P_1, P_2, I)$ matrix class, however in comparison monarch matrices have additional hard constraints on relation between $k_L$ and $k_R$. Being more precise, Monarch matrices are a special case of $\mathcal{GS}(P_1, P_2, I)$ matrices with additional constraints $k_L = b_R^1$, $k_R = b_L^2$. Such constraints lead to several theoretical and practical limitations of Monarch matrices. From theoretical point of view, Monarch matrices can only describe permuted block matrices with blocks of ranks 1. In contrast $\mathcal{GS}$-matrices with can describe matrices with different rank structure of blocks (including structures where rank of each block is equal to arbitrary $r$). From practical point of view, due to this constraint Monarch matrices are often unable to form a desirable block structure of matrices $L$ and $R$. For demonstration of this phenomena, consider a case of square matrices with square blocks – the structure needed in Orthogonal fine-tuning paradigm. Formally, we have $b_L^1 = b_L^2 = b_L$, $b_R^1 = b_2^R = b_R$, $m = n$. Additional Monarch constraint would mean that $b_R = k_L$; $b_L = k_R$. This in turn means that $k_L \cdot k_R = n$. As we can see, it makes impossible to stack two matrices with small number of blocks (say, 4) or large number of blocks, which is required in situations with low parameter budget. In contrast, $\mathcal{GS}$ parametrization allows for both of these structures, which we use in our experiments.

Note, that the work [Fu et al., 2024] provides a slightly different definition for monarch matrices, introducing order-$p$ Monarch matrices. These matrices are also a special case of $\mathcal{GS}$ class, however they are very restrictive as they can only parametrize matrices with both sides equal to $a^p$ for some integers $a, p$.

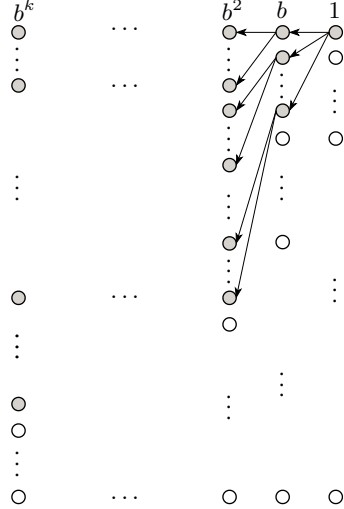

Figure 5: Demonstration of information transition through a block structure. Each node is connected to exactly $b$ consecutive nodes from the next level.

## D   Proof of Theorem 2

We use information transition framework from [Liu et al., 2024b], representing product of $m$ sparse $d \times d$ matrices as an transmitting information in a grid with $d \times (m+1)$ nodes. Edges between nodes $j$ and $i$ represent that element $i, j$ in sparse matrix is non-zero. Element $i, j$ from final matrix can only be non-zero if there exists a path from the $j$-th node from the right column to the $i$-th node in the left column (see Figure 5).

*Proof.* Consider an information transmission graph for the matrix $B_i P_{(r,br)}$. In this graph, the first node connects with $b$ first edges, the second node connects with the edges from $b+1$ to $2b$ and so on. Now consider a graph for the product of $m$ such matrices. As shown in Figure 5, now each node from the first level has paths to $b^k$ unique nodes from the $k$th-th level. It means that using $m = \lceil \log_b(d) \rceil = \lceil \log_b(br) \rceil = 1 + \lceil \log_b(r) \rceil$ matrices is sufficient to reach all nodes and therefore form a dense matrix. Note that the number of paths for each node is always equal to $b^m$ regardless of permutation choice. This observation shows that it is impossible to reach $d$ unique elements on the final level with $m < 1 + \lceil \log_b(r) \rceil$. $\square$

## E   Subject-driven generation

**Training details** All the models are trained using Adam optimizer with batch size = 4, learning rate = 0.00002, betas = (0.9, 0.999) and weight decay = 0.01. The Stable Diffusion-2-base model is used for all experiments.

**Evaluation details** We use the DreamBooth dataset for evaluation. The dataset contains 25 different contextual prompts for 30 various objects including pets, toys and furnishings. For each concept we generate 10 images per contextual prompt and 30 images per base prompt "a photo of an S*", resulting in 780 unique concept-prompt pairs and a total of 8400 images for fair evaluation.

To measure concept fidelity, we use the average pairwise cosine similarity (IS) between CLIP ViTB/32 embeddings of real and generated images as in [Gal et al., 2022]. This means that the image similarity is calculated using only the base prompt, i.e. "a photo of an S*". Higher values of this metric usually indicate better subject fidelity, while keeping this evaluation scene-independent. To evaluate the correspondence between generated images and contextual prompts (TS), the average cosine similarity

between CLIP ViTB/32 embeddings of the prompt and generated images [Ruiz et al., 2023, Gal et al., 2022].

**Overfitting discussion** Typically, the more parameters are trained, the more easily the model overfits. This results in higher image similarity and lower text similarity. In addition, if a model with a large number of training parameters is trained for a long time, the image similarity can often start to decrease as the model starts to collapse and artifacts start to appear. This sometimes leads to models with more trainable parameters having a worse score in both image and text similarity than the models with fewer ones.

Models with fewer trainable parameters overfit less, but require longer training to capture the concept carefully, and usually have an upper limit on the maximum image similarity: at some point the increase in image similarity becomes small, while text similarity starts to decrease dramatically. Therefore, the very common result of a usual fine-tuning is either an overfitting with poor context preservation or an undertraining with poor concept fidelity. The orthogonal fine-tuning shows a different behavior. The model with less trainable parameters can be trained longer (GSOFT, OFT, BOFT) and capture the concept more carefully without artifacts. At the same time, it overfits less and shows higher text similarity.

**Additional results** In Figures 6 we show a graphical representation of the metrics for 1000 and 3000 iterations. Examples of generation for different methods are presented in Figure 7, 8.

# F $\mathcal{GS}$ Orthogonal Convolution

In this section, we provide some details and insights about the choice of the `ChShuffle` permutation and the activation function.

In experiments, we apply the `ChShuffle` operation right before grouped convolutional layers. Stacking several layers of that form resembles higher-order $\mathcal{GS}$-matrices, which motivates the usage of permutations from Definition 5.2 for optimal information transition (see Appendix D). However, in the LipConvnet architecture, the activation function can also shuffle information between channels. Thus, this additional shuffling of information can negatively affect our information transition properties. In the original SOC paper [Singla and Feizi, 2021], the authors use MaxMin activation, firstly proposed in [Anil et al., 2019].

**Definition F.1.** *[Singla and Feizi, 2021] Given a feature tensor $X \in \mathbb{R}^{2m \times n \times n}$, the $MaxMin(X)$ activation of a tensor $X$ is defined as follows:*

$$A = X_{:m,:,:}, \ B = X_{m:,:,:},$$
$$MaxMin(X)_{:m,:,:} = max(A, B),$$
$$MaxMin(X)_{m:,:,:} = min(A, B).$$

This activation shuffles information between different groups in convolution which harms performance of our experiments, as permutations that we use in `ChShuffle` become sub-optimal in terms of information transmission. Thus, we introduce a modification of MaxMin activation, that splits channels into pairs in a different way. Rather than constructing pairs from different halves of input

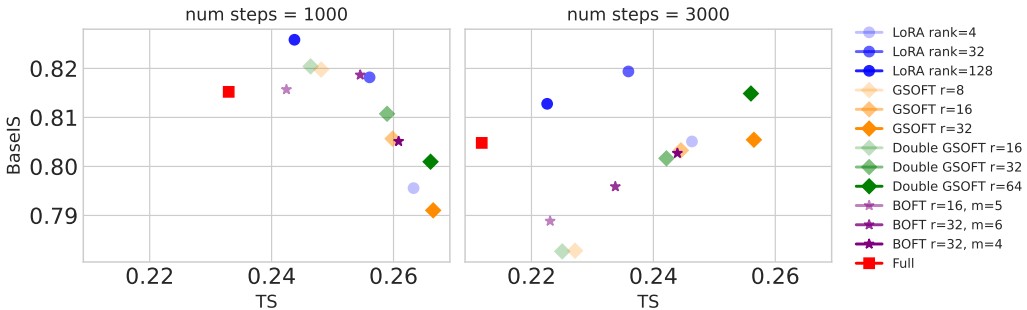

Figure 6: Image and text similarity visualisation for different methods on subject-driven generation.

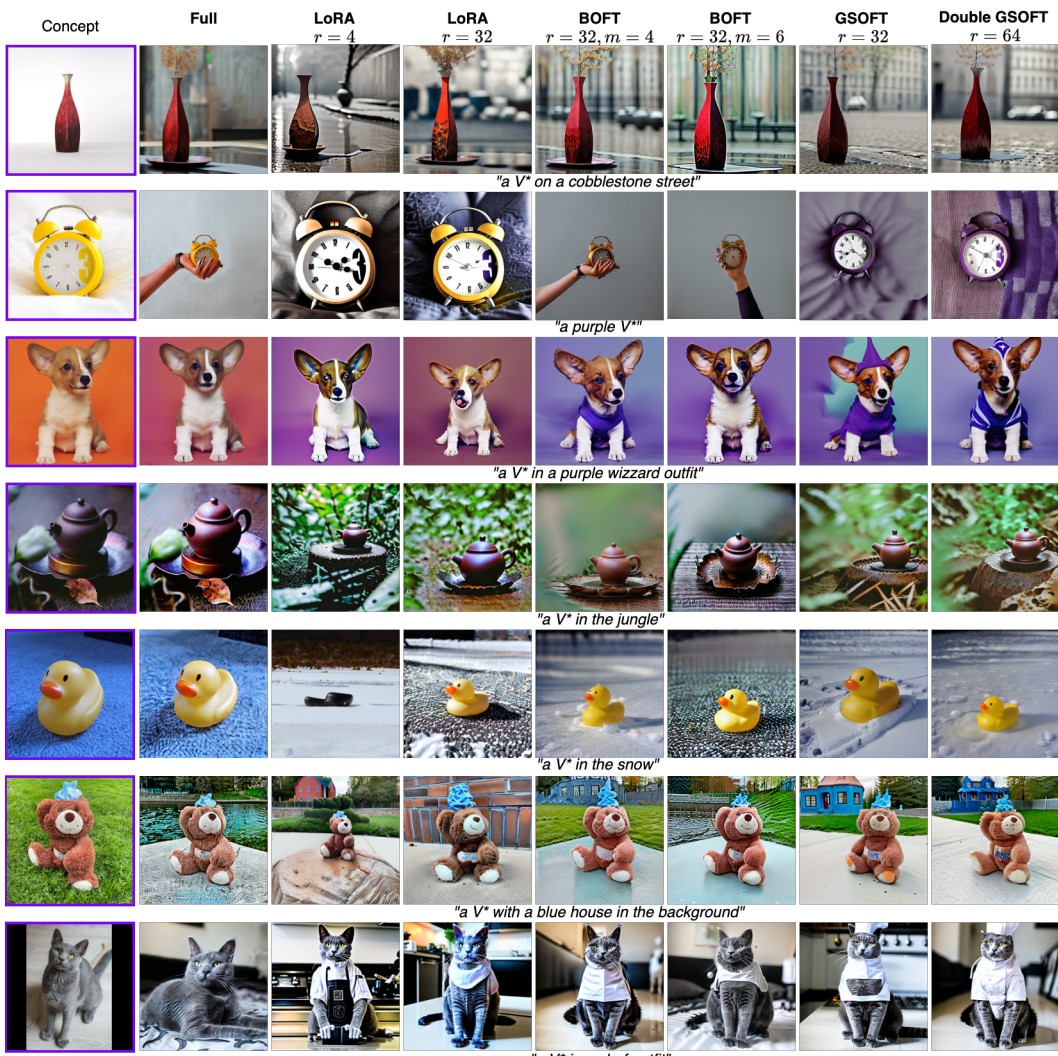

Figure 7: Subject-driven generation visual results on 3000 training iterations.

tensor, we use neighboring channels for forming of pairs (first channel pairs with second, third with fourth and so on). With this modification information does not transfer between groups during activations, which enables more optimal information transmission in-between layers with `ChShuffle` operator. In further experiments we denote this activation function as MaxMinPermuted and define it below:

**Definition F.2.** *Given a feature map* $X \in \mathbb{R}^{2m \times n \times n}$. $MaxMinPermuted(X)$ *is defined as follows:*

$$A = X_{::2,:,:}, B = X_{1::2,:,:},$$

$$MaxMinPermuted(X)_{::2,:,:} = max(A, B),$$

$$MaxMinPermuted(X)_{1::2,:,:} = min(A, B)$$

However, we also empirically find that it is crucial for the channels that interact within activations functions to also interact during convolutions. This means that they should always stay in the same group. This motivates us to use a slightly different permutation for the `ChShuffle` operation, which permutes channels in pairs. We use the following permutation

$$\sigma(i)^{paired}_{(k,n)} = \left( \left\lfloor \frac{i}{2} \right\rfloor \bmod k \right) \cdot \frac{n}{k} + 2 \cdot \left\lfloor \frac{i}{2k} \right\rfloor + (i \bmod 2)$$

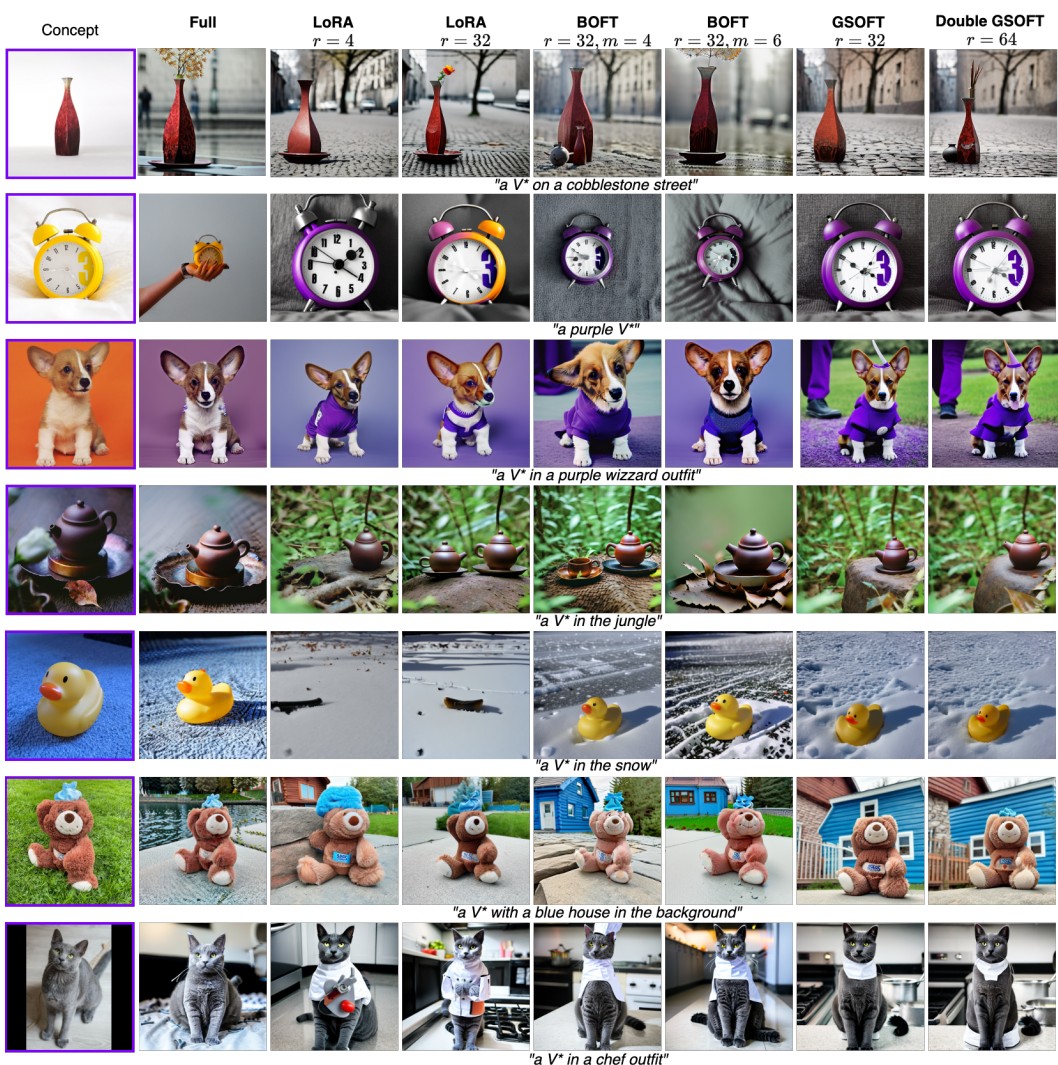

Figure 8: Subject-driven generation visual results on 1000 training iterations.

This permutation can be seen as an adaptation of $P_{(k,n)}$ that operates on pairs of channels instead of single channels. This permutation is also optimal in terms of information transition. We call this permutation "paired". Using this paired permutation as a `ChShuffle` with our modified activation saves connection between pairs while also transmitting information in the most efficient way. We provide the results of comparison of approaches with activations and permutations in Table 4.

Table 4: Comparison of activations on LipConvnet-15 architecture and CIFAR-100. $(a, b)$ in "Groups" column denotes that we have two grouped exponential convolutions (the first one with $kernel\_size = 3$, the second with $kernel\_size = 1$). If $b$ is not mentioned, we have only one $\mathcal{GS}$ orthogonal convolutional layer.

| Conv. Layer | # Params | Groups | Speedup | Activation | Permutation | Accuracy | Robust Accuracy |
|---|---|---|---|---|---|---|---|
| SOC | 24.1M | - | 1 | MaxMin | - | 43.15% | 29.18% |
| GS-SOC | **6.81M** | (4, -) | **1.64** | MaxMinPermuted | paired | **43.48%** | **29.26%** |
| GS-SOC | **6.81M** | (4, -) | **1.64** | MaxMinPermuted | not paired | 40.46% | 26.18% |
| GS-SOC | **6.81M** | (4, -) | **1.64** | MaxMin | paired | 37.99% | 24.19% |
| GS-SOC | **6.81M** | (4, -) | **1.64** | MaxMin | not paired | 39.72% | 25.96% |
| GS-SOC | **8.91M** | (4, 1) | 1.21 | MaxMinPermuted | paired | **43.42%** | **29.56%** |
| GS-SOC | **8.91M** | (4, 1) | 1.21 | MaxMinPermuted | not paired | 40.15% | 26.4% |
| GS-SOC | **8.91M** | (4, 1) | 1.21 | MaxMin | paired | 40.3% | 26.74% |
| GS-SOC | **8.91M** | (4, 1) | 1.21 | MaxMin | not paired | 41.7% | 27.66% |
| GS-SOC | **7.86M** | (4, 2) | 1.22 | MaxMinPermuted | paired | **42.86%** | **28.98%** |
| GS-SOC | **7.86M** | (4, 2) | 1.22 | MaxMinPermuted | not paired | 41.13% | 27.53% |
| GS-SOC | **7.86M** | (4, 2) | 1.22 | MaxMin | paired | 41.55% | 27.45% |
| GS-SOC | **7.86M** | (4, 2) | 1.22 | MaxMin | not paired | 41.25% | 27.29% |
| GS-SOC | **7.3M** | (4, 4) | 1.23 | MaxMinPermuted | paired | **42.75%** | **28.7%** |
| GS-SOC | **7.3M** | (4, 4) | 1.23 | MaxMinPermuted | not paired | 38.93% | 25.59% |
| GS-SOC | **7.3M** | (4, 4) | 1.23 | MaxMin | paired | 40.34% | 27.06% |
| GS-SOC | **7.3M** | (4, 4) | 1.23 | MaxMin | not paired | 41.57% | 27.48% |

It can be seen that using "paired" permutation used with MinMaxPermuted activation significantly improves quality metrics.

