# OpenReview forum: "Group and Shuffle: Efficient Structured Orthogonal Parametrization"
_NeurIPS.cc/2024/Conference — NeurIPS 2024 poster_

### Official Review · Reviewer_Kbzt · 2024-07-11

**Soundness:** 3
**Presentation:** 3
**Contribution:** 3
**Rating:** 5
**Confidence:** 4

**Summary:**

In this paper, the authors proposed a new structured matrices, namely GS-matrices. It unifies and generalizes structured classes from previous works. Compared to the previous BOFT, it further improves the parameters and training efficiency. Experiments on language processing, text-image generation, and image classification have demonstrated the performance of GSOFT.

**Strengths:**

1. The writing is easy to understand.
2. The proposed GSOFT has been applied to various networks on various downstream tasks.

**Weaknesses:**

1. It's better to provide the training time comparison between BOFT, LoRA, and GSOFT. It seems that some results are lost in Table 2.
2. It's better to discuss more about single GSOFT and double GSOFT. It seems that the results in Table 1&3 are from the single version and the results in Table 2 are from the double version.
3. In Table 2, it seems that more learnable parameters result in lower CLIP scores. Maybe more discussions are required.

**Questions:**

My main concerns are listed in the weaknesses.
1. It's better to demonstrate the performance of the proposed method on recent popular GPT series methods (e.g., Llama2, Llama3, Mixtral). This may expand the influence of this method in the field.

---

> ### Author Rebuttal · Authors · 2024-08-06
>
> **Training times and double GSOFT.** Thank you for raising these questions. We have fixed these issues in the top-level comment.
>
>
> **CLIP scores.** Thank you for pointing out this observation. Typically, the more parameters are trained the more easily the model overfits. This results in higher image similarity and lower text similarity. In addition, if a model with a large number of training parameters is trained for a long time, the image similarity can often start to decrease as the model starts to collapse and artefacts start to appear.
>
> Models with fewer trainable parameters overfit less, but require longer training to capture the concept carefully, and usually have an upper limit on the maximum image similarity: at some point the increase in image similarity becomes small, while text similarity starts to decrease dramatically. Therefore, the very common result of a usual fine-tuning is either an overfitting with poor context preservation or an undertraining with poor concept fidelity.
>
> The orthogonal fine-tuning shows a different behaviour. The model with less trainable parameters can be trained longer (OFT, BOFT) and capture the concept more carefully without artefacts. At the same time, it overfits less and shows higher text similarity.
>
> Indeed, this needs to be discussed and we will add this to the paper, thank you.
>
> **GPT series methods.** While we certainly agree with your point, RoBERTa experiment took us multiple weeks of compute time. Conducting experiments with multi-billion models would require the amount of computational resources that we unfortunately do not currently have. Nevertheless, we believe that our experiments are of interest, as they show the comparison with the closest competitor BOFT on a different domain apart from diffusion experiments.

---

> > ### Comment · Reviewer_Kbzt · 2024-08-13
> > **Thanks for Response**
> >
> > Thanks for the authors' response. My main concerns have been addressed. Thus, I will keep my score.

---

> > > ### Author Response · Authors · 2024-08-14
> > >
> > > We would like to thank the reviewer for the response!

---

### Official Review · Reviewer_yzRC · 2024-07-12

**Soundness:** 2
**Presentation:** 3
**Contribution:** 2
**Rating:** 4
**Confidence:** 3

**Summary:**

This paper follows the orthogonal fine-tuning paradigm that uses orthogonal matrices for adapting the weights of a pretrained model. It introduces a class of structure matrices, defined as GS-matrices, that can well be used to conduct orthogonal matrix. Based on GS-matrices, It proposes structured orthogonal parametrization and studies its performance in the orthogonal fine-tuning framework. Experiments are conducted on two standard setting ups: downstream task fine-tuning in language modeling and text-to-image diffusion tasks. It also claims the proposed method can be used to construct orthogonal convolutions for 1-Lipschitz neural networks.

**Strengths:**

The description of this paper is very clear, and the overall presentation is good. It clearly describes how this paper motivates from orthogonal fine-turning. The proposed method using structured matrix to conduct orthogonal matrix is interesting.

**Weaknesses:**

1.The experiments are not well conducted to support the claims/effectiveness of the proposed method:
(1) This paper spends many words to claim the efficiency of the proposed methods. However, this paper does not provide the experiments to show the wall-clock time in the natural language understanding and subject-driven generation. I think this paper should provide the wall-clock time when comparing to baselines in Table 1 and Table 2, to show the efficiency of the proposed methods, compared to the baselines. What is the results of wall-clock time of these methods in Table 1 and 2?
(2) This paper should conduct some ablation study to the Double GSOFT. It seems that Double GSOFT is the key to improve the performance, based on the description of this paper, but the double idea is apparently incremental. Besides, why Table 1 show the results using GSOFT, while Table 2 using Double GSOFT? Why not provide both in the experiments or using the best one (maybe Double GSOFT) with ablation study? I think this paper should address this point in the rebuttal.

2.This paper claims that “Nevertheless, parametrization of orthogonal matrices is a challenging task” In Line 19. I do not think so. There are many previous solutions to conduct orthogonal matrix in DNNs before 2019[1,2,3,4], even for conducting orthogonal convolutions [5,6]. And there are merits and drawbacks among these methods. I think this paper should well discuss these previous methods and take credits to them.

3.The original motivation of this paper is for parametrization efficient finetuning by using orthogonal fine-tuning. Meanwhile, it also wants to claim its contributions to the application orthogonal matrix benefits, E.g., conducting 1-Lipschitz network for robustness.
But this claim is not well supported by experiments (There are many other orthogonal convolution methods(e.g., [5] and [6]), not only the baselines in the paper. E.g. SOC). Besides, this paper should pay more attention on discussing the orthogonal techniques in DNNS during the related work section.


**Ref:**
[1] On orthogonality and learning recurrent networks with long term dependencies, ICML 2017
[2] Learning unitary operators with help from u(n), AAAI 2017
[3] Orthogonal recurrent neural networks with scaled Cayley transform, ICML, 2018
[4] Orthogonal weight normalization: Solution to optimization over multiple dependent stiefel manifolds in deep neural networks, AAAI 2018
[5] Preventing Gradient Attenuation in Lipschitz Constrained Convolutional Networks, NeurIPS 2019
[6] LOT: Layer-wise Orthogonal Training on Improving ℓ2 Certified Robustness, NeurIPS 2022

**Questions:**

The questions in weakness 1

**Limitations:**

Yes

---

> ### Author Rebuttal · Authors · 2024-08-06
>
> **W1.** Thank you for your comments. We added compute times and comparison to the standard GSOFT for subject-driven generation (see the top-level comment). In the NLP tasks, we are also faster or on par with the BOFT approach. Compared to LoRa we are several times slower, but observe a slight quality boost on the scale we consider. The BOFT paper shows more quality gain for larger models like Llama. Unfortunately, such experiments require much more computational resources.
>
> **W2 and W3.** Thank you for pointing out these papers. We will be sure to make a more detailed survey in the revised version.
> To be more specific, [1]-[3] work in the setting of dense matrices and try to parameterize essentially the whole manifold of orthogonal matrices. For $n \times n$ matrices this requires computing inverses or exponential maps of $n \times n$ matrices at every optimization step. This takes $O(n^3)$ time, which can be computationally challenging for larger architectures. Moreover, such a parametrization utilizes $\mathcal{O}(n^2)$ trainable parameters, which makes it inapplicable for PEFT (see the OFT paper [Qiu et. al, NeurIPS '23] for more details). Our method is different in a sense that it provides a trade-off between expressivity (describing only a subset of the orthogonal manifold) and efficiency. For example, for a dense $n \times n$ matrix, when setting a number of blocks equal to $\sqrt{n}$, our method uses $O(n \sqrt{n})$ parameters and does $O(n^2 \sqrt{n})$ computations at optimization steps.
>
> Work [4] applies orthogonal parametrization to a reshaped convolution tensor.  Although it is indeed a widely used heuristics, it does not formally lead to an orthogonal convolution mapping and certified robustness. The work [5] was outperformed by the SOC method that we utilize (there is a comparison in the SOC paper). Finally, in [6], the authors utilize periodicity of convolution and their padding is not equivalent to a widely-used zero-padded convolution. What is more, the survey [7] suggests that [6] and other 1-Lipschitz architectures yield worse robustness metrics than SOC, which is why it serves as a baseline in our paper.
>
> [7] B. Prach, et al. "1-Lipschitz Layers Compared: Memory Speed and Certifiable Robustness." Proceedings of the IEEE/CVF Conference on Computer Vision and Pattern Recognition. 2024.

---

> > ### Author Response · Authors · 2024-08-14
> >
> > Please let us know if you have any follow-up questions that need further clarification. Your insights are valuable to us, and we stand ready to provide any additional information that might be helpful.

---

### Official Review · Reviewer_AkW1 · 2024-07-12

**Soundness:** 3
**Presentation:** 3
**Contribution:** 2
**Rating:** 4
**Confidence:** 3

**Summary:**

This paper introduces a new way of constructing orthogonal parametrization for efficient model fine-tuning. The development of the proposed method is motivated by: naive orthogonal fine-tuning, which can be too restrictive due to how it constructs the block-diagonal orthogonal matrix; and improved methods with dense orthogonal matrix can be too expensive in terms of both computation and parameters.
The authors introduce a new way of constructing orthogonal matrix by an alternating product of block-diagonal matrices and several permutations, with advantages over previous methods justified theoretically.
Experiments on language model, stable diffusion, and standard conv models are presented.

**Strengths:**

- The proposed method is able to significantly reduce the parameter and computation compared to the previous method on dense orthogonal parameter tuning.

- The overall writing of this paper is good. And the presentation is well organized.

- The core advantages of the proposed methods are justified with theoretical supports. The development of GS Orthogonal Convolutions makes it more generalizable to more applications.

**Weaknesses:**

My major concern with this paper is the empirical justifications.

**Small experiment scale.**
All experiments are conducted in very small scale. Specifically, the language models used in this paper are relatively old and outdated; Only few-shot personalization experiments are reported in the stable diffusion experiments; and the final conv application is only conducted on a tiny network on cifar100.
This makes it very hard to objectively evaluate the proposed method's practical effectiveness.

**Missing evaluations**
If I'm not missing anything, the entire *training time* row of Table 2 is empty. Training time and memory are the core advantages compared to the previous methods. While it is true that the method is able to reduce the parameters and computations by numbers, significant speed-up and memory reduction in practice can still be a huge support to the paper's contribution.

**Missing comparisons**
The comparisons reported in this paper are mainly on LoRA and other orthogonal tuning methods. However, without the comparisons against other improved LoRA methods, such as DoRA, it can be hard to position the contribution and impact of this work.

**Questions:**

- Does the SVD need to be performed in every optimization step?

**Limitations:**

The limitation discussions presented in Section 9 focus mainly on how the model compares to LoRA and BOFT. The proposed method still requires more parameters than other improved LoRA methods.

---

> ### Author Rebuttal · Authors · 2024-08-06
>
> **Small experiment scale.** We do understand your concerns regarding the scale of experiments.
>
> For diffusion experiments, the few-shot setting is one of the most challenging tasks for fine-tuning. On the one hand, the model should carefully capture the target concept, but on the other hand, it should preserve it's ability to follow the target prompt. This setting is one of the main motivations for orthogonal fine-tuning (OFT), as it additionally helps to avoid overfitting and was used in the pioneering work with the OFT paradigm.
>
> Despite the few shot setting of a task, we report results accumulated over **30 different concepts and 25 contextual text prompts**. In total, there are 750 unique concept-prompt pairs and **a total of 7500 images for each baseline for robust evaluation**. And our evaluation includes different hyperparameters for each baseline method.
>
> Notice that BOFT only presents a visual comparison on $11$ different concepts with one fixed set of hyperparameters and does not provide a qualitative evaluation. Thus, our evaluation in this task is more comprehensive and gives more insight into how different models perform with different parameters.
>
> For 1-Lipschitz architectures, we took a standard neural network that was specifically developed for smaller datasets. We are not aware of any other architecture baselines that guarantee exact 1-Lipschitzness and are developed for larger datasets such as ImageNet. The works we aware of do not go beyond smaller datasets in their experiments, see e.g. [1, 2, 3]. One of the factors for this absence could be the high computational complexity associated with 1-Lipschtiz layers.
>
> Finally, regarding the NLP experiment, the main concern for us was the computational resources required for running larger models. RoBERTa experiment took us multiple weeks of compute time. Conducting experiments with multi-billion models would require the amount of computational resources that we unfortunately do not have. Nevertheless, we believe that our experiments are of interest, as they show the comparison with the closest competitor BOFT on a different domain apart from diffusion experiments.
>
> **Missing evaluations.** Thank you for pointing this out. We added training times for the Table 2 (see the top-level comment). Both GSOFT and Double GSOFT are faster (and better in terms of metrics) than BOFT.
>
> **Missing comparisons.** Thank you for this comment. We did an additional experiment with DoRA for subject-driven generation. We found that DoRA gives slightly better results compared to LoRA, but still shows significantly lower text similarity compared to GSOFT and DoubleGSOFT. **See Table 1 and Figure 1 from attached PDF file.** We also plan to add DoRA comparison for NLP experiment in the revisited version.
>
> **SVD during optimization steps**. We actually did not use SVD at any step of the process. The SVD steps we describe when projecting to the class of GS matrices can be a useful tool for future applications. For example, for different initiazation strategies of GS-matrices.
>
> [1] Singla et al. "Skew orthogonal convolutions." International Conference on Machine Learning. 2021
>
> [2] M. Laurent, et al. "A dynamical system perspective for lipschitz neural networks." International Conference on Machine Learning. 2022
>
> [3] X. Xiaojun, et al. "Lot: Layer-wise orthogonal training on improving l2 certified robustness." Advances in Neural Information Processing Systems. 2022

---

> > ### Author Response · Authors · 2024-08-14
> >
> > Please let us know if you have any follow-up questions that need further clarification. Your insights are valuable to us, and we stand ready to provide any additional information that might be helpful.

---

### Official Review · Reviewer_dcw2 · 2024-07-29

**Soundness:** 3
**Presentation:** 3
**Contribution:** 3
**Rating:** 7
**Confidence:** 4

**Summary:**

This paper proposes a new parameterization for orthogonal matrices and applies this parameterization to orthogonal finetuning of foundation models. The new orthogonal parameterization is interesting and useful. The experimental results show some improvement over the baselines.

**Strengths:**

- The study of parameter-efficient orthogonal finetuning is very much in need and is an important topic at the mement. I believe this new parameterization will be useful for orthogonal finetuning in general.

- The GS parameterization seems novel to me, although I am not well knowledgeable about the related orthogonal parameterization literature. However, in the application of orthogonal finetuning, the paper, to my knowledge, is the first work to apply this parameterization.

- Specifcially, the GS parameterization resembles the Monarch matrices, as mentioned in the paper. To some extend, it can be viewed as a generalization of the Monarch matrices.

- The experimental results look promising and demonstrate that its parameter-efficiency outperforms both OFT and BOFT in some scenarios.

**Weaknesses:**

- Although it is trivial to show that GS matrices can approximate arbitrary orthogonal matrices (as it generalizes the Monarch matrices), it is still important to discuss and (potentially re-use) the theoretical results from the Monarch matrices.

- It can further strengthen the paper, if the authors can add the comparison of the Monarch parameterization in the orthogonal finetuning experiments.

- The implementation should be released prior to its acceptance, otherwise it may be difficult for people to reproduce the results.

**Questions:**

See the weakness section.

**Limitations:**

The major concern that I has is whether the results are easily reproducible. With that being said, the authors should consider to release the source code for the orthogonal finetuning experiments.

---

> ### Author Rebuttal · Authors · 2024-08-06
>
> Thank you for your comments and suggestions. First of all, we add a source code for our experiments (see the top-level comment). Regarding the Monarch matrices, the key drawback in the Monarch representation for us is that if one block-diagonal factor has many small blocks, then the other one has to have several large dense blocks. As a result, we are not able to obtain as few parameters as we want, which was critical for our experiments. For example, in the natural language understanding experiment, obtaining number of parameters similar to LoRa with rank 8 is impossible with Monarch Matrices, which would require at least 4x parameters with any block structure. On the other hand, additional flexibility introduced in GS matrices allows for matrices with small number of blocks in each of the matrices, making it possible to use this method with any parameter budget.
> At the same time, the order-p generalization of Monarch matrices from the follow-up paper, is only capable of working with very specific matrix sizes (see also Appx. C). What is more, the used permutation matrices do not allow for optimally filling the matrix with non-zeroes using a given budget of parameters.
> Following your suggestions, we will emphasize these points in the revised version and add more discussion concerning the Monarch matrix class.

---

> > ### Comment · Reviewer_dcw2 · 2024-08-10
> > **Thanks for the response**
> >
> > My concerns are properly addressed. I think this paper proposes an interesting parameterization for orthogonal matrices and is worth discussing at the conference. I will keep my current score and recommend for acceptance.

---

> > > ### Author Response · Authors · 2024-08-14
> > >
> > > We would like to thank the reviewer for the response!

---

### Author Rebuttal · Authors · 2024-08-06

We thank the reviewers for taking the time to review our article.
We appreciate your feedback and the constructive remarks you have provided.
Before giving detailed answers to the raised questions, we briefly reiterate the main contribution of our paper.
We propose a new class of matrices, denoted as $\mathcal{GS}$ that generalizes previous matrix classes: Monarch matrices, their order-p generalization, and Block Butterfly matrices.
We theoretically show how to choose permutations in the $\mathcal{GS}$-class to address the shortcomings of the previous approaches.
We showcase our class on multiple application domains, including text-to-image diffusion models, language models and convolutional architectures.

In addition to the specific answers below, we address several key common questions from the reviewers in the top-level comment:

1.  We add an anonymised version of our code to ensure the reproducibility of the results (Reviewer dcw2). Anonymous link to the code was provided to AC in a separate comment following NeurIPS guidelines.

 2.  We extended the results in Table 2 for the whole Dreambooth dataset, that contains $30$ concepts of different categories, including pets, interior decoration, toys, backpacks, etc. For each concept, we used $25$ contextual text prompts, which include accessorisation, appearance and background modification. We also added DoRa baseline (Reviewer AkW1). As an ablation study, we conduct experiments in a non-doubled version of GSOFT (Reviewers yzRC, Kbzt). Both GSOFT and Double GSOFT show similar performance in terms of text similarity, outperforming other baselines with the same number of training parameters. However, as the number of training parameters decreases, Double GSOFT ($r=64$) provides better image similarity and shows the most balanced solution in terms of concept fidelity and context preservation among all baselines. **See Table 1 from attached PDF file.**

 3. We provide training times for the Table 2 (Reviewers AkW1, yzRC, and Kbzt). Both GSOFT and Double GSOFT are faster (and better in terms of metrics) than BOFT and comparable to DoRA. **See Table 1 from attached PDF file.**

4. We also provide more visual comparison for subject-driven generation to illustrate the performance of the baselines on different prompts and concepts. **See Figure 1 from attached PDF file.**

---

### Decision · Program_Chairs · 2024-09-25

**Decision:**

Accept (poster)

**Comment:**

The paper introduces a novel structured matrix class, GS-matrices, which unifies and extends existing classes like Monarch and Block Butterfly matrices. The proposed approach is applied to orthogonal finetuning, showing parameter efficiency across different tasks, including text-to-image diffusion models and language modeling. Reviewer dcw2 praised the novelty and practical impact of the GS parameterization, rating it as an accept due to the strong potential for broader application in orthogonal finetuning. However, the reviewer suggested incorporating comparisons with Monarch parameterization and releasing the implementation for reproducibility, which the authors addressed by providing the code and discussing the limitations of Monarch matrices.

Reviewer AkW1 highlighted the theoretical contributions and the method’s ability to reduce parameters and computation, but expressed concerns over the small scale of experiments, lack of training time data, and missing comparisons with newer LoRA methods like DoRA. The authors responded by adding training times and conducting additional comparisons, yet AkW1 maintained a borderline reject rating due to perceived limitations in empirical validation.

Reviewer yzRC found the presentation clear but questioned the efficiency claims without wall-clock time data and suggested ablation studies for Double GSOFT. They also noted the need for a broader discussion on previous orthogonal matrix parameterization methods. The authors responded with additional data and a more detailed survey of related work, but yzRC still rated the paper as borderline reject, citing incremental contributions and limited experimental validation.

Reviewer Kbzt recognized the broad application of GSOFT but recommended additional training time comparisons and broader evaluations on recent models like Llama2. Although the authors provided further clarifications and data, Kbzt did not raise their score, keeping it at borderline accept, indicating that while their concerns were addressed, broader experimental evidence was still needed.

Overall, the paper presents a promising and technically solid contribution to orthogonal finetuning. However, the concerns regarding empirical validation and broader impact, as raised by multiple reviewers, suggest that further refinement and additional experiments would strengthen the paper. Given the addressed feedback and the potential impact, I recommend acceptance, with an emphasis on expanding the experimental evaluations in future work to fully realize the method’s contributions.